# Experiments Studying the Instability Process of a Subway Tunnel in Soil–Rock Composite Strata Influenced by Defects

Ruichuan Zhao [1], Yunfei Zheng [2], Yongjian Guo [3], Shaoshun Luan [2] and Sulei Zhang [2,*]

[1] CCCC HIGHWAY CONSULTANTS Co., Ltd., Beijing 100010, China; zhaoruichuan2004@163.com
[2] School of Civil Engineering, Qingdao University of Technology, Qingdao 266033, China; Z1641902123@163.com (Y.Z.)
[3] Qingdao Conson Second Jiaozhou Bay Subsea Tunnel Co., Ltd., Qingdao 266071, China
* Correspondence: zhangsulei@qut.edu.cn

**Abstract:** Subway tunnels excavated in soil–rock composite strata face great challenges due to the prevalence of inner defects. The instability of tunnels in these strata poses significant risks to construction safety. In this paper, indoor experiments are adopted to study the instability process of a subway tunnel in soil–rock composite strata influenced by inner cavities. A total of six groups of tests are designed based on the location of the cavity and the distance of the cavity from the tunnel. High-resolution monitoring techniques are employed to capture the real-time deformation and failure process of strata. The results show that a cavity in the strata significantly affects the stability of the strata after the tunnel excavation. The existence of a cavity increases ground deformation, and a cavity at different locations affects ground deformation. The strata around the cavity are the first to experience failure by the upper loads applied after the tunnel is excavated. The location of the cavity changes the stress distribution state of the strata and thus alters the emergence of cracks, which finally disturbs the collapse process and pattern of the composite strata. The probability and collapse range increase when the tunnel excavation impact zone is connected with the cavity weakening zone. The findings can provide technical support for the collapse prevention and safety control of subway tunnels in composite strata with internal defects.

**Keywords:** soil–rock composite strata; inner defect; instability process; ground deformation; indoor experiment

## 1. Introduction

Urban underground development has intensified globally, driven by rapid population growth and the demand for sustainable transportation systems. Subway tunnels excavated in soil–rock composite strata face great challenges due to the inherent heterogeneity of geological materials and the prevalence of natural or inner defects. Composite strata, characterized by alternating layers of soil and rock, exhibit abrupt transitions in mechanical properties, leading to stress concentration and differential deformation under external loads [1,2]. Such geological complexity is further exacerbated by defects, including voids, fractures, weak interlayers, and material discontinuities, which act as stress concentrators and pathways for groundwater infiltration [3–5]. The instability of tunnels in these strata poses significant risks, including localized collapse, water inrush, and structural failure, threatening construction safety (Figure 1). The urgency of addressing these challenges is underscored by increasing urbanization. However, existing design methodologies often inadequately account for defect-induced instability mechanisms, relying on simplified

homogeneous models. This gap has prompted researchers to investigate composite strata behavior through experimental and numerical approaches, aiming to refine risk assessment frameworks and mitigation strategies.

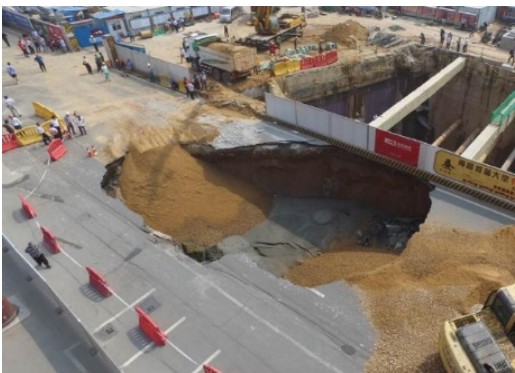 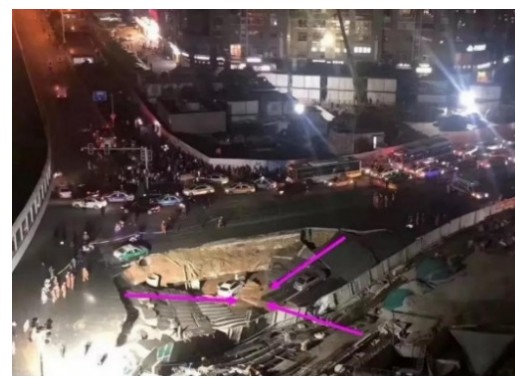

**Figure 1.** Strata failure caused by metro tunnel construction.

Experimental investigations have been pivotal in elucidating failure mechanisms of subway tunnels in composite strata. Physical model tests, scaled to replicate field conditions, enable researchers to observe stress redistribution, deformation patterns, and progressive collapse under controlled scenarios. For example, Yan et al. [3] investigated a catastrophic collapse during subway excavation in Qingdao, China, using centrifuge tests to simulate the groundwater-induced softening of sandy cobble layers. They identified a critical threshold of pore pressure beyond which seepage erosion triggered rapid cavity formation and tunnel face instability. Advanced monitoring techniques, such as distributed fiber optic sensing (DFOS) and 3D digital image correlation (DIC), have enhanced experimental precision. Li et al. [6] employed DFOS to track strain localization in tunnel linings under asymmetric loading, demonstrating that defect clusters near the tunnel crown amplified bending moments by up to 40%. Meanwhile, Wei et al. [7] utilized DIC to quantify crack propagation in scaled tunnel models, correlating defect density with collapse energy thresholds. These studies collectively highlight the sensitivity of composite strata to defect distribution and hydrological conditions, underscoring the need for defect-specific reinforcement strategies.

Defects in composite strata significantly alter mechanical responses, necessitating targeted experimental studies [4,5]. Cavities and fractures, whether pre-existing or induced by excavation, disrupt stress arching and reduce load-bearing capacity. Physical model tests by Carranza-Torres and Reich [8] demonstrated that circular cavities in cohesive soil–rock interfaces reduced tunnel face stability by 25–30%, with failure modes transitioning from shear-dominated to tensile-dominated as cavity size increased. Similarly, Senent and Jimenez [9] designed a series of trapdoor tests to simulate partial collapses in layered strata, showing that defect clustering within 1.5 times the tunnel diameter increased surface settlement by 150%. Di et al. [10] conducted seepage experiments on sandy cobble strata, revealing that high groundwater flow rates (exceeding 0.1 m/s) eroded defect boundaries, forming preferential seepage channels that accelerated pore pressure dissipation. Their results aligned with field observations from the Guangzhou Metro, where water-rich fractured zones required grouting pressures 20% higher than theoretical predictions to stabilize tunnel faces. Additionally, Huang et al. [11] developed a fluid–solid coupling apparatus to simulate shield tunneling under high water pressure, identifying a nonlinear relationship between defect orientation and seepage force magnitude. Vertically aligned defects amplified hydraulic gradients, while horizontal defects promoted lateral soil loosening. Conversely, Rostami [12] analyzed crushed zones around disk cutters in mixed-face tunneling, demonstrating that rock proportion exceeding 70% suppressed defect propagation but increased cutter wear. Li et al. [13] conducted a series of experiments to investigate

the failure mechanisms of rock blocks with combined flaws under coupled static and dynamic loads. Chen et al. [14] investigated failure mechanisms of cavity-induced railway tunnel collapse in defect phyllite strata, proposing steel arch replacement and grouting reinforcement to enhance rock lining structural integrity. Shiau et al. [15] investigated the ground stability of ellipsoidal cavities due to pipeline defects using advanced finite element limit analysis with adaptive meshing in an axisymmetric condition. Zhou et al. [16] investigated the coupling defect effect of elliptical cavities and cracks on tunnel dynamic stability and failure modes through AUTODYN-based numerical simulations and modified drop-hammer impact tests. It can be seen that inner defects can disrupt the stress state and reduce the load-bearing capacity of strata.

Subway tunnels excavated in composite strata face great challenges due to the inherent heterogeneity. Research has also been performed in this regard. For example, Zhang et al. [1] investigated the collapse mechanism of subway tunnels in soil–sand–rock composite strata through physical model tests. They highlighted the impact of stratum heterogeneity on tunnel stability and identified key failure modes under different loading conditions. Liu et al. [2] conducted model tests to investigate the progressive collapse mechanism of shallow subway tunnels in soft upper and hard lower composite strata. They revealed that the collapse process is influenced by the interaction between the soft and hard layers, with stress redistribution playing a critical role in the failure pattern. The findings emphasized the role of stress anisotropy in controlling the failure patterns and collapse progression. It can be concluded that the composite strata significantly influences tunnel stability and collapse mechanisms. These studies provide valuable insights into the collapse mechanisms of tunnels in composite strata, highlighting the importance of considering stratum heterogeneity in tunnel risk assessment.

It can be found that both ground defects and non-homogeneous characteristics of composite strata can affect the stability of tunneling projects. However, most of the existing studies on ground defects are focused on a unique stratum, and the few studies on composite strata have failed to systematically reveal the effects of different locations of cavities on tunnel stability. Existing experimental studies have significantly advanced the understanding of tunnel failures in composite strata, yet critical gaps persist. Therefore, the collapse mechanism of soil–rock composite strata influenced by the inner cavity should be figured out. To address these limitations, this study adopts indoor experiments to study the instability process of a subway tunnel in soil–rock composite strata influenced by inner cavities. A total of six groups of tests were designed based on the location of the cavity and the distance of the cavity from the tunnel. High-resolution monitoring techniques are employed to capture the real-time deformation and failure process of strata. Finally, a predictive framework for the instability risk assessment is used and we propose targeted reinforcement measures, such as defect-specific grouting protocols.

## 2. Model Test Design

### 2.1. Similarity Theory of Model Test

Similarity theory is a doctrine used to study the principle of similar phenomena in engineering, and it is the basis for model test research, which can satisfy the similarity theory to make the model test and the actual phenomenon have the same physical parameters and physical relationships. According to the results of previous research, the sufficient conditions for similarity of physical phenomena can be summed up in the following three theorems:

(1)    First theorem of similarity

This theorem was first proposed by Newton in 1688, and then a rigorous proof was given by the French scholar Beltrán in 1848. The first theorem of similarity can be expressed

as follows: if the phenomena are similar and the single-valued conditions are the same, then the similarity index is equal to 1 or the similarity criteria are equal, and the single-valued conditions include geometric conditions, physical conditions, boundary conditions, and initial conditions.

(2)  Second theorem of similarity

This theorem was proposed and derived by the Russian scholar Fetelman in 1911. The second theorem of similarity can be expressed as follows: if the phenomena are similar, then the physical equations describing the similar phenomena can be transformed into a functional relationship between the similarity criteria, and the functional relationship between the similarity criteria is the same.

(3)  Third theorem of similarity

This theorem was proposed by the former Soviet scholar Kirpichov in 1930. The third theorem of similarity can be expressed as follows: where the same phenomena are similar if the single-valued conditions are similar and the similarity criterion consisting of such single-valued conditions is numerically equal, then these phenomena must be similar.

Similarity theory dictates that the equilibrium equations, geometric compatibility conditions, constitutive relations, stress boundaries, and displacement boundaries must satisfy strict congruence between the prototype and scaled models [9–11]. The similarity ratios of the model test have the following relation:

$$\begin{cases} C_\sigma = C_L C_\gamma \\ C_\delta = C_L C_\varepsilon \\ C_\sigma = C_E C_\varepsilon \end{cases} \tag{1}$$

where $C_\sigma$, $C_L$, $C_\gamma$, $C_\delta$, $C_\varepsilon$, and $C_E$ represent the similarity ratio of stress, geometry, density, displacement, strain, and elastic modulus, respectively. Taking into account the basic conditions of the model test and the various influencing factors in the test, the similarity of the model test selected for the model test of soil–rock composite strata containing voids in this paper is as follows: the similarity ratio of geometric $C_l = 20$, similarity ratio of stress $C_\sigma = 20$, similarity ratio of elastic modulus $C_E = 20$, similarity ratio of displacement $C_u = 20$ and similarity ratio of cohesion $C_c = 20$; however, the similarity ratio of gravity $C_\gamma = 1$, similarity ratio of strain $C_\varepsilon = 1$, similarity ratio of Poisson's ratio $C_\mu = 1$, and similarity ratio of friction angle $C_\varphi = 1$.

### 2.2. Model Test Setup

The model test device mainly includes three parts, a model test box, loading system, and control system, as shown in Figure 2. This test uses an airbag unloading system for tunnel excavation simulation and a non-contact video monitoring system for monitoring changes at various points within the surrounding rock and ground settlement. The size of the model test box is 2.0 m × 0.3 m × 1.8 m (length × width × height). To easily observe the changes in the stratum inside the box during the loading process, a plexiglass panel that can withstand a certain amount of pressure is used on the front of the box, with a 300 mm diameter hole in the center of the panel, which is the same as the diameter of the tunnel model. To reduce the boundary effect, the center of the circular hole was located 60 cm away from the lower boundary of the front panel and 85 cm away from the left and right boundaries. To increase the rigidity of the front and back of the box, columns are added to the front and back panels of the box. The columns are connected to the base of the box with screws and connected to the top beam track plate so that the overall structure of the box is strengthened. To add soil material, we disengage the connecting plate of the top beam of the box. A roller mounted on the top beam can be pushed to the rear of the test box to complete the filling. The loading system consists of servo-hydraulic cylinder

steady pressure and steady flow loading. The automatic control system consists of PLC control and a human–machine interface touchscreen. To simulate the real tunnel excavation process, a non-elastic cylindrical airbag is buried at the tunnel location of the model box, which is connected to the air compressor through a pressure valve, and the pressure value inside the airbag is controlled through the pressure valve. During the test, the pressure is released step by step through a pressure valve to simulate the tunnel excavation process. After the deformation of the stratum is stabilized, the airbag is pulled out. A non-contact video monitoring system is used to monitor and record the changes in the internal points of the surrounding rock and the settlement of the strata.

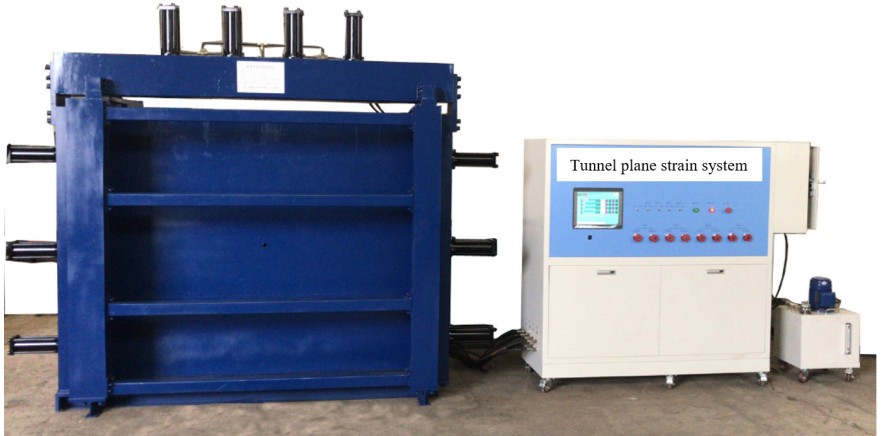

**Figure 2.** Model test equipment, control systems, and loading systems.

*2.3. Similarity Materials*

To highlight the difference between soil and rock layers, the soil layer is considered as Class VI surrounding rock, and the rock layer is considered as Class IV surrounding rock. According to the 1:20 similarity ratio of the model test, the parameters of the corresponding materials were converted. A corresponding model with similar materials to the prototype was prepared. The conversion relationship between the physical and mechanical indexes of the prototype and the model of the two types of surrounding rock is shown in Table 1.

**Table 1.** Physical–mechanical parameters of Class IV and ClassVI surrounding rock .

| Surrounding Rock | / | Gravity, $\gamma$ (KN/m³) | Elastic Modulus, $E$ (GPa) | Poisson's Ratio, $\mu$ | Friction Angle, $\varphi$ (°) | Cohesion, $c$ (MPa) |
|---|---|---|---|---|---|---|
| IV | Prototype | 20–23 | 1.3–6 | 0.3–0.35 | 27–39 | 0.2–0.7 |
| | Model Test | 20–23 | 0.052–0.24 | 0.3–0.35 | 27–39 | 0.008–0.028 |
| VI | Prototype | 15–17 | <1 | 0.4–0.45 | <20 | <0.2 |
| | Model Test | 15–17 | <1 | 0.4–0.45 | <20 | <0.2 |

In addition to the selection of similar materials to meet the basic physical and mechanical properties of the test indicators, we generally need to agree on the following principles: similar materials should have similar physical and mechanical properties with the prototype material; that is, the material should be uniform and isotropic. Changing the ratio of the material can make its physical and mechanical properties meet the test requirements and can be deployed to different physical and mechanical parameters of the model material. The performance of the molded material has a certain amount of stability and is not affected by external conditions (temperature, humidity, etc.) and less affected

by the outside world. The material itself is reasonably priced, and the system is easy to transport and process for production.

Gypsum-based materials, sand/paraffin/resin, and barite powder/bentonite blends are similar materials that are often used, and 3D-printed composites are currently more popular. However, all of the above materials have advantages and limitations and need to be employed according to the research objectives and requirements. Considering the properties, ease of access, and economics of similar materials, barite, quartz sand, and petroleum jelly were selected in conjunction with established research studies. Despite the limitations mentioned above, model tests are a very common research tool today. They have significant advantages in representing the real mechanical state and failure process of the strata. In this paper, the material mixed with quartz sand and barite as the aggregate and petroleum jelly as the binder is chosen as the model test material. This material can not only meet the above requirement but also has the characteristics of a short forming time, a high capacity of mixed material, and stable mechanical properties, and it is reusable. Combined with the previous research results, this paper adopts a similar material ratio of barite/quartz sand/petroleum jelly = 10:4.8:1 for class IV surrounding rock and barite/quartz sand/petroleum jelly = 8:5:0.6 for class VI surrounding rock. The physical and mechanical parameters of similar materials are shown in Table 2.

**Table 2.** Physical and mechanical parameters of similar materials.

| Material | Gravity, $\gamma$ (KN/m$^3$) | Elastic Modulus, $E$ (GPa) | Poisson's Ratio, $\mu$ | Internal Friction Angle, $\varphi$ (°) | Cohesion, $c$ (MPa) |
|---|---|---|---|---|---|
| IV | 20 | 1.3–6 | 0.3 | 27–39 | 0.2–0.7 |
| VI | 15 | <1 | 0.4 | <20 | <0.2 |

*2.4. Model Test Schemes*

The test mainly investigates the stability of the ground during the excavation of tunnels through the ground with cavities. According to the relative position and distance between the cavity and the tunnel, a total of six tests were identified. Based on the geometric similarity ratio of 1:20, the tunnel diameter is determined to be 30 cm, and the tunnel depth is 60 cm. The rock-to-span ratio is 1:3, where the thickness of the upper soil layer is 50 cm. The diameter of the cavity is 10 cm. The cavities are designed to be located directly above the tunnel, diagonally above the tunnel, and at the foot of the tunnel. The distance between the cavity and the tunnel is designed to be 10 cm, 20 cm, and 30 cm. The specific test plan is shown in Table 3, and a schematic diagram of the model test plan is presented in Figure 3.

**Table 3.** Model test plan.

| Test | Buried Depth of Tunnel/cm | Rock-to-Span Ratio | Tunnel Diameter/cm | Cavity Diameter/cm | Distance Between Cavity and Tunnel/cm | Cavity Location |
|---|---|---|---|---|---|---|
| 1 | 60 | 1:3 | 30 | 10 | / | / |
| 2 | 60 | 1:3 | 30 | 10 | 10 | right above |
| 3 | 60 | 1:3 | 30 | 10 | 10 | oblique upper part |
| 4 | 60 | 1:3 | 30 | 10 | 10 | lower part of side |
| 5 | 60 | 1:3 | 30 | 10 | 20 | oblique upper part |
| 6 | 60 | 1:3 | 30 | 10 | 30 | oblique upper part |

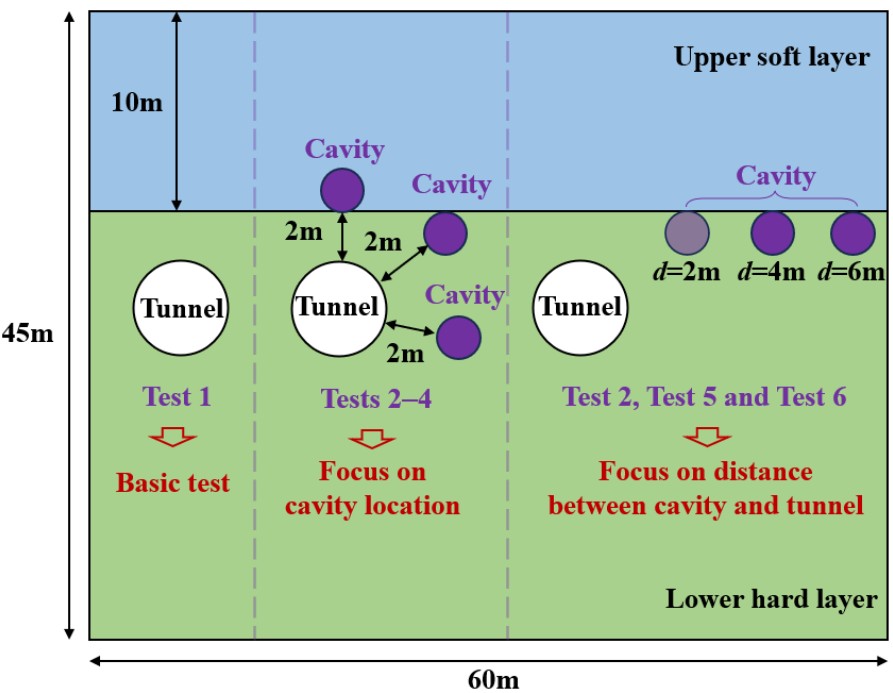

**Figure 3.** A schematic diagram of the model test plan.

The scale of the model test is 1:20, the tunnel diameter is 30 cm, and the tunnel center is 100 cm away from the model boundary. Therefore, the influence of the boundary effect in this experiment is not significant. In addition, to further weaken the boundary effect, the bottom and both sides of the test box were filled with cushioning material with similar stiffness to the soil body. Moreover, this experiment is a staged application of load at the top of the model conducted to avoid the superposition of stress waves reflected at the boundary due to transient loading.

*2.5. Model Test Procedure*

The main procedure of the mode test is as follows:

(1) Material configuration process: Firstly, weigh the materials required for the test according to the decided ratio. Preheat the quartz sand and barite powder in the oven at 90–100 degrees Celsius, and melt the petroleum jelly by heating it in a water bath. After the preparation work has been carried out, mix and color the three test materials by manual and mechanical mixing.

(2) Material and tunnel model filling and cavity pre-embedding: Similar materials are filled in the test model stand. Control the height of the material pile and the weight of the filled material by manual tamping to control the density of the soil body. Obtain the strength of similar materials in line with the requirements of the model test. Fill the soil to the bottom of the tunnel model and the location of the cavity, pre-bury the tunnel airbag to the location of the tunnel model and pre-bury the airbag in the cavity. After the similar material is filled and compacted by applying static loads, deflate the airbags to form ground cavities and leave them to rest for one hour.

(3) Tunnel airbag unloading: The airbag is gradually unloaded by controlling the pressure value to simulate the stress relief during tunnel excavation. After unloading the tunnel airbag completely, the airbag is pulled out to simulate tunnel excavation. Although the airbag may not replicate natural void geometries, it has been proven that the airbag is a simple and effective method to simulate cavities. This simplified method can also better represent the changes in the strata stability caused by cavities [17–19].

(4) Application of upper load: After the deformation of the stratum is stabilized, the upper load is applied by the test loading system step by step until the deformation of the stratum is destroyed.

(5) Data acquisition: A non-contact video monitoring system is used to record the results of surface settlement, as well as the damage pattern of the surrounding rock during the deformation and destruction of the strata after the application of upper loads.

## 3. Model Test Results

### 3.1. Test Results for Strata Without Defects

Test 1 is a tunnel excavation in a non-defective stratum, which serves as a control test. Figure 4 shows the surface settlement curve above the tunnel when the tunnel is excavated and left to stand for one hour without the application of upper loads. It can be seen that the maximum surface settlement value of 2.51 mm occurs at the centerline of the tunnel, i.e., at the surface directly above the tunnel.

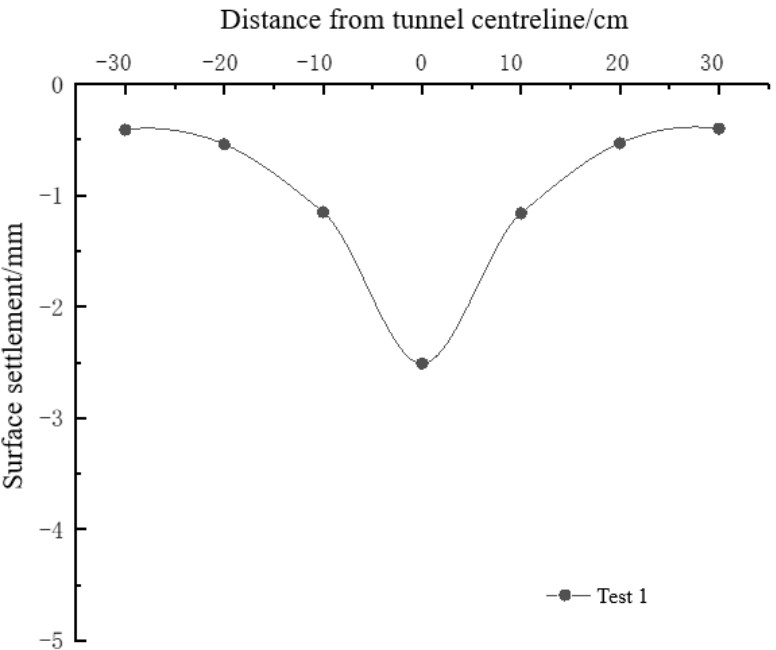

**Figure 4.** Surface settlement curve for stratum without defects.

Figure 5 shows the destabilization and failure process of the strata when upper loads are applied after tunnel excavation. It can be seen that no significant changes are observed in the tunnel at the initial stage of load application. With the increase in the upper load, the tunnel arch shoulder and the arch top show an obvious falling block phenomenon (Figure 5c). When the upper load continues to increase, the rock layer at the top of the tunnel moves downwards, and shear crack appears at the shoulder of the tunnel. Then, failure first occurs at the tunnel shoulder. With the increase in the load, shear cracks are constantly present, and a progressive collapse happens in the lower rock layer, as shown in Figure 5d. When the collapse expands to the soil–rock interface, tensile cracks are observed at the soil layer under the increasing load. Therefore, a collapse quickly occurs in the upper soil layer. Considering the lower load-bearing capacity of the soil layer, the collapse speed of the overburdened layer is accelerated. When the loading value is stable at about 10 kPa, the overburdened layer completely collapses to a stable state, forming a collapse-through-type failure. Several collapse examples of actual engineering are selected to compare the final collapse pattern of the composite strata (Figure 6). It can be found that the fracture

surface of the strata in both the model test and the actual project is larger the closer it is to the subsurface, where the final collapse patterns of both are the same.

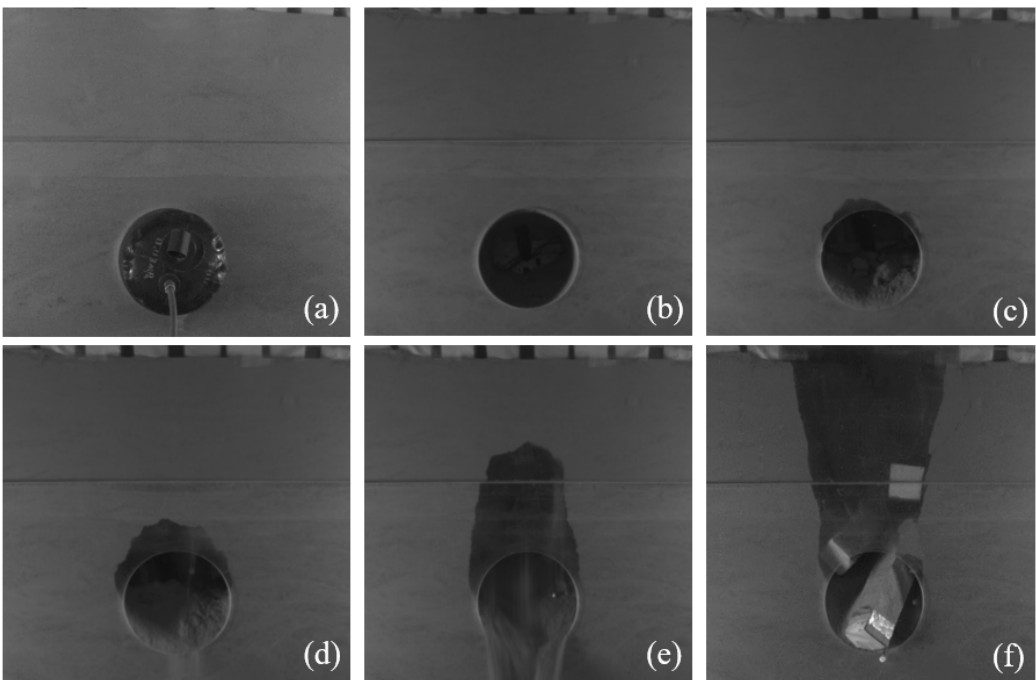

**Figure 5.** Progressive collapse of composite strata without defects. (**a**) Initial state. (**b**) Tunnel excavation. (**c**) Collapse occurs. (**d**) Collapse expands in rock layer. (**e**) Collapse expands to soil layer. (**f**) Finally, collapse pattern.

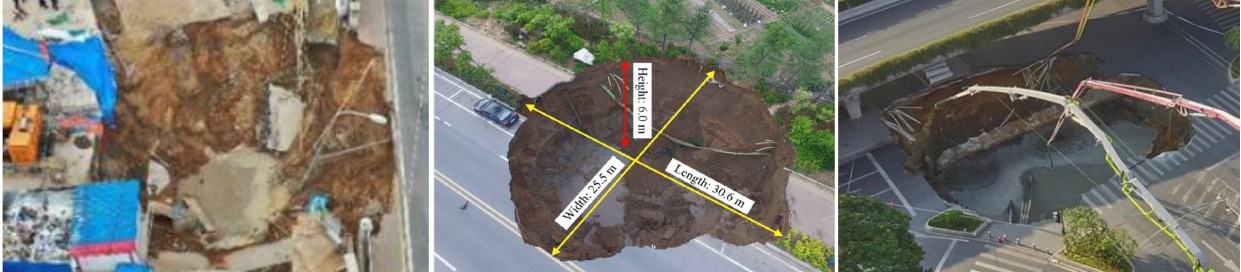

**Figure 6.** Collapse pattern of composite strata [2].

*3.2. Test Results for Strata with Defects*

3.2.1. Influence of Cavity Location

Surface settlement curves for strata without defects in different locations are illustrated in Figure 7. The cavity in Test 2 is located directly above the tunnel, and the maximum surface settlement value of 3.14 mm is found at the surface directly above the tunnel, which is symmetrically distributed due to the occurrence of the cavity and the center of the tunnel. In Test 3, the cavity is located diagonally above the tunnel, and the maximum surface settlement value of 3.42 mm can be obtained from the right side of the surface directly above the tunnel. As the center of the cavity and the tunnel do not meet in Case 3, the settlement law of the strata also changes, and the maximum settlement position is shifted to the side of the cavity. The cavity in Test 4 is located on the outer side of the tunnel foot, and the maximum surface settlement value of 3.64 mm is also shifted to the side of the cavity. The maximum surface settlement is observed when the cavity is located on the outer side of the tunnel foot. It can be seen that the presence of strata cavities can change the stability of the strata and subsequently have an effect on the surface deformation pattern.

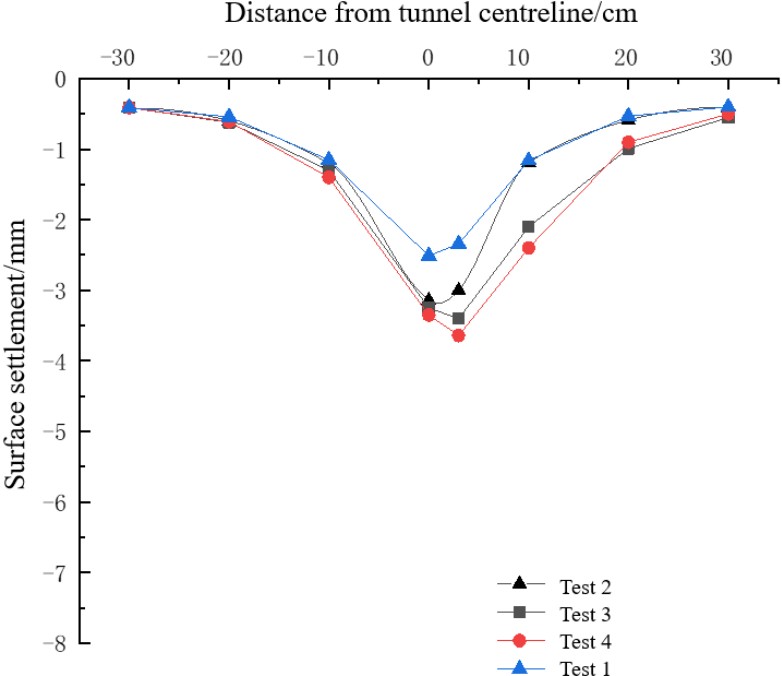

**Figure 7.** Surface settlement curve for stratum without defects in different locations.

Figure 8 shows the failure process of strata when upper loads are applied after tunnel excavation in the presence of a cavity directly above the tunnel. From Figure 8b, it can be seen that there is no significant change in the cavity and tunnel when it is not loaded. As shown in Figure 8c, a distinct vertical crack extends upward from the left arch shoulder. This is a shear crack in the arch shoulder under external loads and excavation. The crack continues to propagate upward under the external load. As the load increases, this crack connects with the upper cavity. Subsequently, a rupture surface is formed in the upper left part of the tunnel, which is followed by a collapse, as shown in Figure 8d. When the load is increased to 6 kPa, the stratum collapse develops to the overlying soil layer. Similarly, as the collapse extends into the upper soil layer, tension cracks continue to appear under external loading. The rate of collapse extension is increased, taking into account the low bearing capacity of the soil layer. At this time, the upper load is stopped, and the soil layer collapse still develops rapidly up to the ground surface.

Figure 9 shows the failure process of strata when the cavity is located diagonally above the tunnel. As can be seen from Figure 9b, some falling is observed at the tunnel vault as soon as the excavation of the tunnel is completed, and the falling increases significantly when the upper load is applied. A tension crack appears at the location of the right arch shoulder of the tunnel because the void on the right side changes the stress distribution of the strata (Figure 9c). As the load increases, the expansion of the tension cracks reduces the stability of the rock formation on the right side, and thus, the arch collapses. It can be noticed that this collapsed area is towards the cavity side when the load is 4 kPa (Figure 9d). At this stage, the ground falls away severely and the rupture surface connects the tunnel to the cavity. Then, the collapse of the soil layer still develops violently after stopping the application of the upper load. In addition, new cracks have appeared around the cavity due to stress concentrations. Therefore, the overlying soil layer on the side close to the cavity is the first to collapse under the action of self-gravity stress (Figure 9e). The final failure pattern of the strata shows that the area of collapse is significantly higher on the side close to the cavity than on the side away from the cavity (Figure 9f).

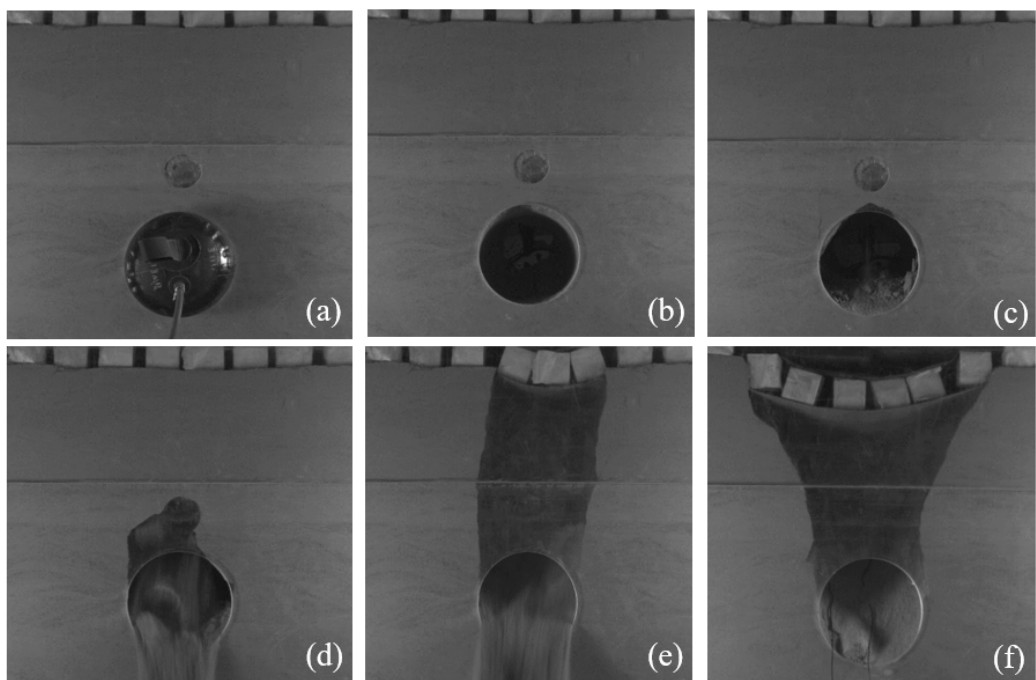

**Figure 8.** Progressive collapse of composite strata with cavity above tunnel. (**a**) Initial state. (**b**) Tunnel excavation. (**c**) Collapse occurs. (**d**) Collapse expands in rock layer. (**e**) Collapse expands to soil layer. (**f**) Finally, collapse pattern.

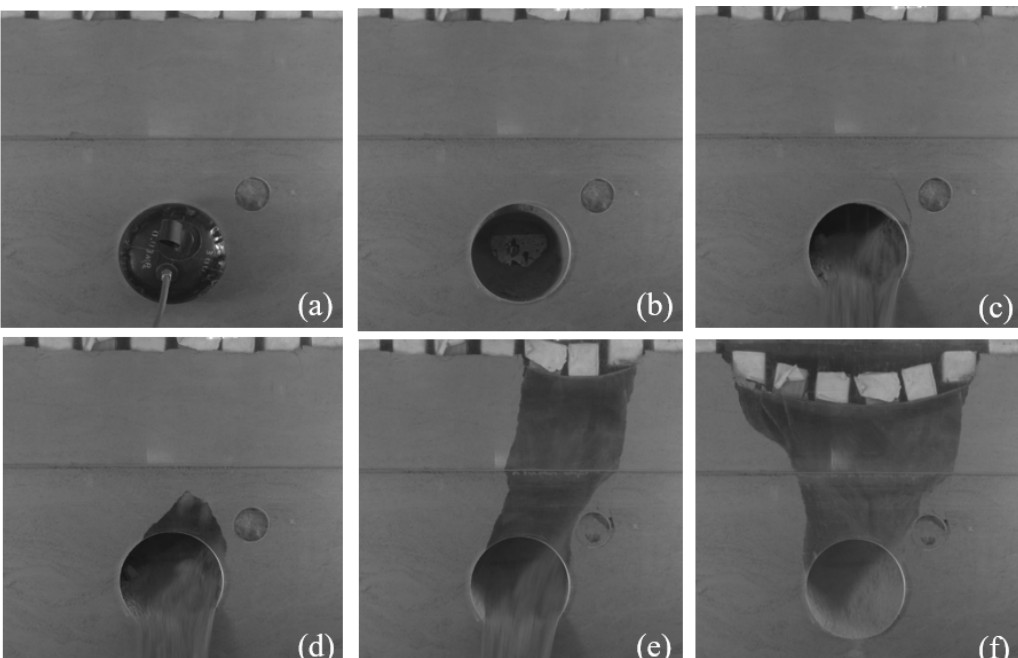

**Figure 9.** Progressive collapse of composite strata with cavity at tunnel shoulder. (**a**) Initial state. (**b**) Tunnel excavation. (**c**) Collapse occurs. (**d**) Collapse expands in rock layer. (**e**) Collapse expands to soil layer. (**f**) Finally, collapse pattern.

Figure 10 shows the failure process of strata when the cavity is located on the outside of the tunnel foot. As can be seen in Figure 10b, when the tunnel excavation is completed, shear cracks appear above the tunnel and the cavity. As the load increases, shear cracks appear first on the left side. Immediately, the failure appears on the left side wall of the tunnel. The shear cracks on the left side continued to expand upwards under increasing loads. At this time, the right side cracks also appear. When the cracks on both sides expand

at the same time, the bearing capacity of the rock layer on the upper side of the tunnel arch is reduced, so the collapse occurs (Figure 10d). However, we found that no significant collapse occurred at the location of the cavity due to its distance. When the load value goes up to 4 kPa, the soil layer collapses drastically and rapidly collapses above the ground surface, forming a collapse-through type of damage (Figure 10e). From the final failure pattern of the strata in Figure 10f, it can be seen that the severe failure of the strata and the largest amount of strata collapse occur when the cavity is located at the foot of the tunnel arch. In addition, the surface above the side near the cavity is the first to collapse and the amount of collapse is much larger than the side away from the cavity.

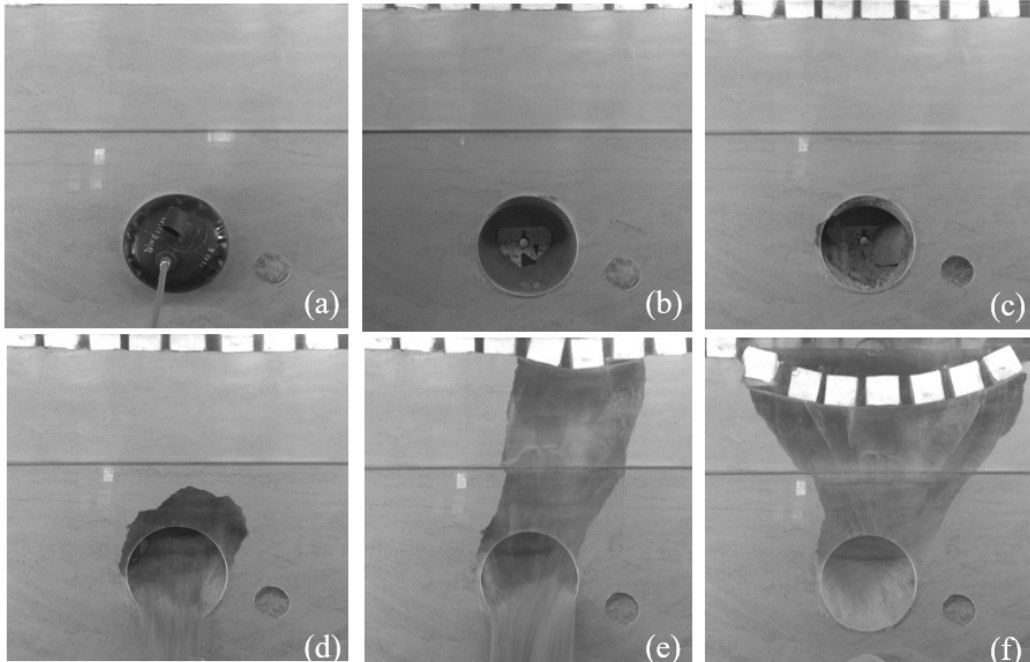

**Figure 10.** Progressive collapse of composite strata with cavity at tunnel foot. (**a**) Initial state. (**b**) Tunnel excavation. (**c**) Collapse occurs. (**d**) Collapse expands in rock layer. (**e**) Collapse expands to the soil layer. (**f**) Finally, collapse pattern.

A comparison of the surface settlement and the failure process of the strata caused by the tunnel excavation at different locations shows that the maximum surface settlement is greater and the damage to the strata is more serious when the cavern is located on the outer side of the tunnel foot compared to the other locations, followed by the diagonal upper part of the tunnel, and the smallest deformation of the strata is found when the cavern is located directly above the tunnel.

### 3.2.2. Influence of Distance Between Tunnel and Cavity

Figure 11 gives the surface settlement curves for different distances of the cavity on the outer shoulder of the tunnel. The maximum ground settlement of the tunnel is 3.42 mm, 3.16 mm, and 3.0 mm when the distance of the tunnel from the cavity is 2 m, 4 m, and 6 m, respectively. The further the cavity is from the tunnel, the smaller the ground settlement is, which indicates that the closer the cavity is to the tunnel, the bigger the influence on the stability of the strata.

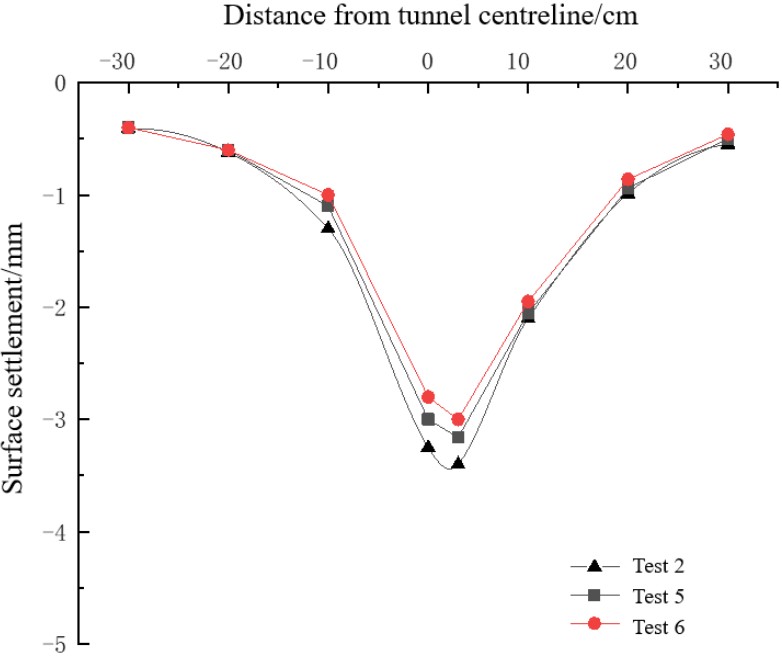

**Figure 11.** Surface settlement curve for stratum influenced by distance between tunnel and cavity.

Figure 12 shows the progressive collapse of composite strata when the cavity is located diagonally above the tunnel and the clear distance between the tunnel and the cavity is 4 m. No significant cracking is observed in the vicinity of the cavity as the load increases (Figure 12b). With the application of the upper loads, the strata above the tunnel chamber start to drop blocks (Figure 12c). Similarly, tension cracks appeared first in the sidewalls on both sides of the tunnel under continuously increasing loads. The cracks extend upwards, leading to a collapse above the tunnel arch (Figure 12d). As can be seen from Figure 12d, the collapse pattern is symmetrically distributed. This shows that the cavity has less influence on the strata within the excavation disturbance at this stage. At this point, the upper load is stopped, and the overlying soil layer collapses to the ground surface under the action of self-gravitational stress. In this process, the strata on the side close to the cavity are still the first to collapse. From the final damage pattern of the strata (Figure 12f), the collapse zone of the strata near the cavity is larger than that away from the cavity. This also suggests that the presence of a cavity affects the collapse process and morphology of the composite strata.

Figure 13 shows the collapse process of strata instability when the cavity is located diagonally above the tunnel and the clear distance between the tunnel and the cavity is 6 m. From Figure 13b, it can be seen that when the tunnel excavation is completed without loading, there is no significant change inside the tunnel. However, as the load increases, cracks appear equally on both side walls of the tunnel. Thus, the strata around the tunnel appear to collapse when the upper load is applied up to 4 kPa (Figure 13c). During this stage, cracks continue to expand, leading to a progressive collapse within the rock layer. When the upper load is continuously applied to 8 kPa, the rock layer above the tunnel collapses completely and the overlying soil layer begins to collapse (Figure 13d). At this point, the overlying soil layer collapses to the surface under self-gravitational stresses after the application of the load ceases (Figure 13e). From Figure 13f, the final damage pattern of the strata, it can be seen that when the net distance between the cave and the tunnel reaches 6m, the influence of the strata cavities on the stability of the strata is small.

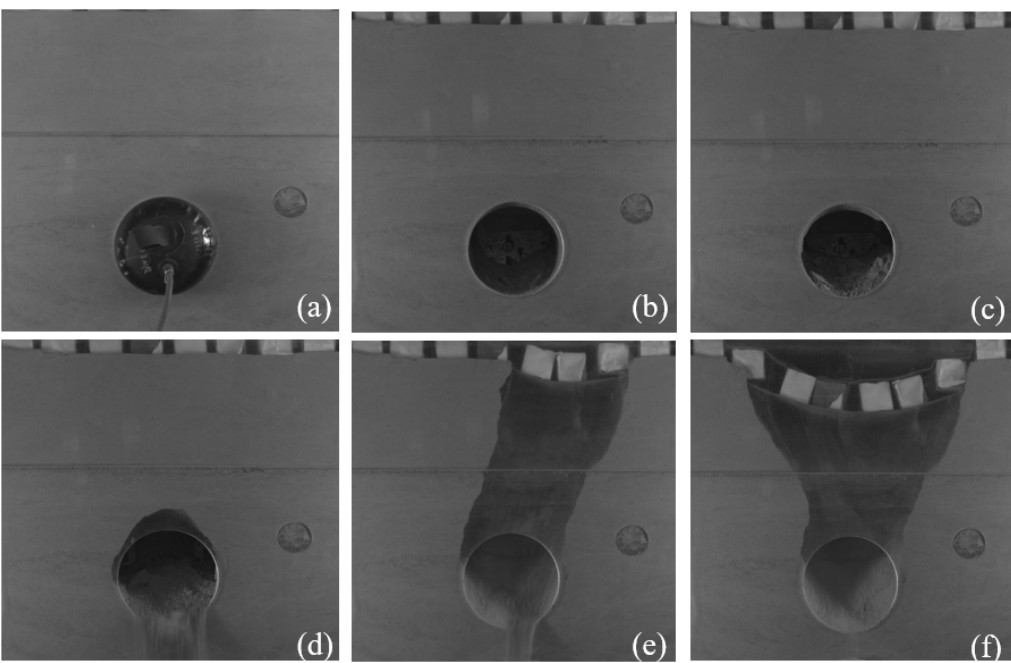

**Figure 12.** Progressive collapse of composite strata with cavity at tunnel shoulder at distance of 4 m. (**a**) Initial state. (**b**) Tunnel excavation. (**c**) Collapse occurs. (**d**) Collapse expands in rock layer. (**e**) Collapse expands to soil layer. (**f**) Finally, collapse pattern.

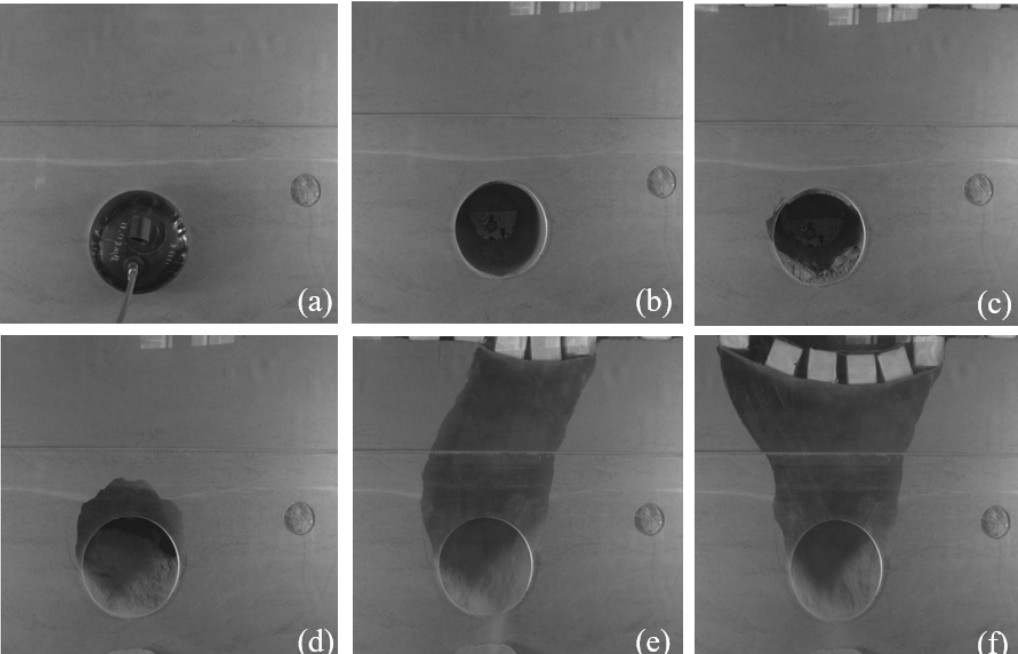

**Figure 13.** Progressive collapse of composite strata with cavity at tunnel shoulder at distance of 6 m. (**a**) Initial state. (**b**) Tunnel excavation. (**c**) Collapse occurs. (**d**) Collapse expands in rock layer. (**e**) Collapse expands to soil layer. (**f**) Finally, collapse pattern.

## 4. Discussion

The presence of cavities can affect the stability of the strata during tunneling. Figure 14 shows the maximum surface settlement excavation induced and ultimate loads for strata influenced by cavity location. It can be seen that when there is no cavity, the maximum deformation of the ground surface caused by tunnel excavation is 2.51 mm and the ultimate bearing capacity of the strata is 10 kPa. The maximum deformation of the ground surface caused by tunnel excavation in Case 2, Case 3, and Case 4 are 3.14 mm, 3.42 mm, and

3.64 mm, respectively. The results showed that the existence of a cavity would increase the ground deformation, and the cavity at different locations has an effect on the ground deformation. In addition, the loads of strata failure in Case 2, Case 3, and Case 4 are 6 kPa, 4 kPa, and 4 kPa, respectively. It is seen that the existence of cavities significantly reduces the stability of the strata, and the failure occurs in advance under the external loads.

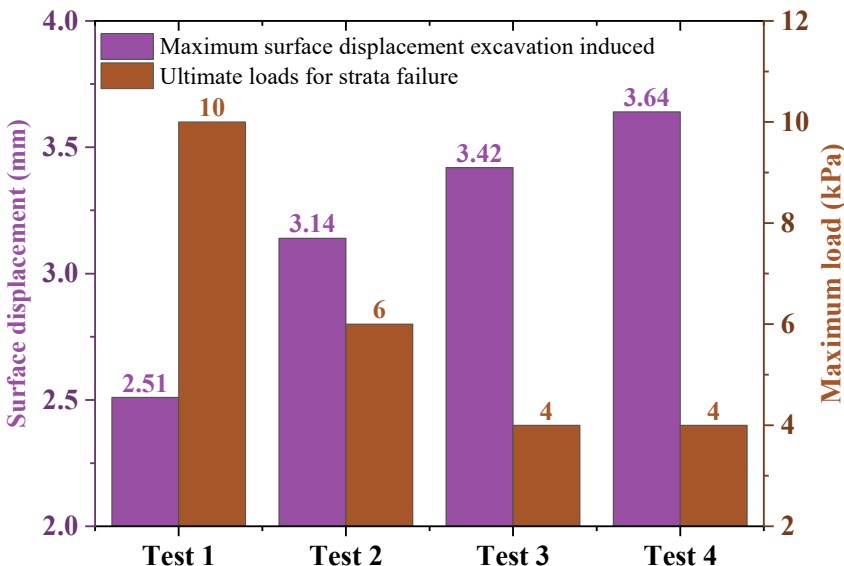

**Figure 14.** Maximum surface settlement excavation induced and ultimate loads for strata influenced by cavity location.

In addition, the effect of the distance between the cavity and tunnel on the strata stability is investigated. The deformation and bearing capacity of the strata for various distances between the cavity and the strata are given in Figure 15. When the distance between the tunnel and the cavity is 2 m, 4 m, and 6 m, respectively, the maximum deformation of the ground surface is 3.42 mm, 3.16 mm, and 3 mm, and the bearing capacity is 4 kPa, 6 kPa, and 8 kPa, respectively. With the increase in the distance between the tunnel and the cavity, the maximum surface settlement value decreases accordingly, and the intensity of the strata failure is also reduced accordingly, which indicates that the further the tunnel excavation is away from the cavity of the ground layer, the safer it is.

It can be seen that the existence of ground cavities greatly affects the safety of tunnel construction. When the tunnel excavation impact zone is connected with the cavity weakening zone, the probability and range of stratum collapse will increase, which is the reason why many accidents happen nowadays. Therefore, to avoid similar accidents, it is necessary to take advanced measures for reinforcement treatment. One solution is given in Figure 16, which is to reinforce the strata by grouting around the tunnel. This grouting reinforcement must extend beyond the zone affected by the tunnel excavation to reduce the probability of interaction between the cavity and the tunnel, thus improving the safety of the strata. The following commonly used common grouting materials include cement slurry, cement–water–glass bi-liquid slurry, chemical slurry, and ultra-fine cement slurry. The material chosen needs to be decided according to the strata type. The grouting pressure is usually 0.5–3.0 MPa, depending on the formation conditions and the depth of the grouting. The initial grouting pressure is low and gradually increases to the design pressure. The effectiveness and reasonableness of the grouting material and grouting pressure can be validated by indoor tests and field tests. Therefore, through a reasonable selection of materials and processes, grouting reinforcement can effectively improve the stability and safety of subway tunnels.

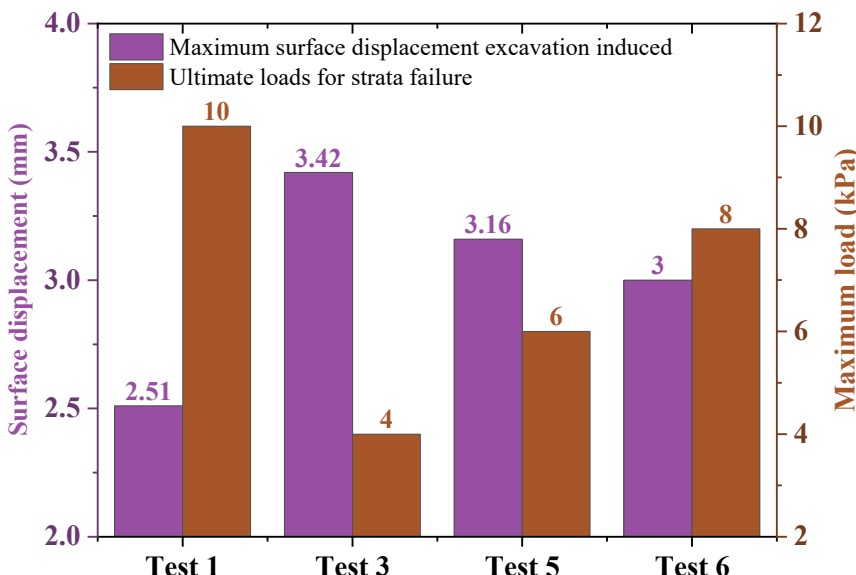

**Figure 15.** Maximum surface settlement excavation induced and ultimate loads for strata influenced by cavity distance.

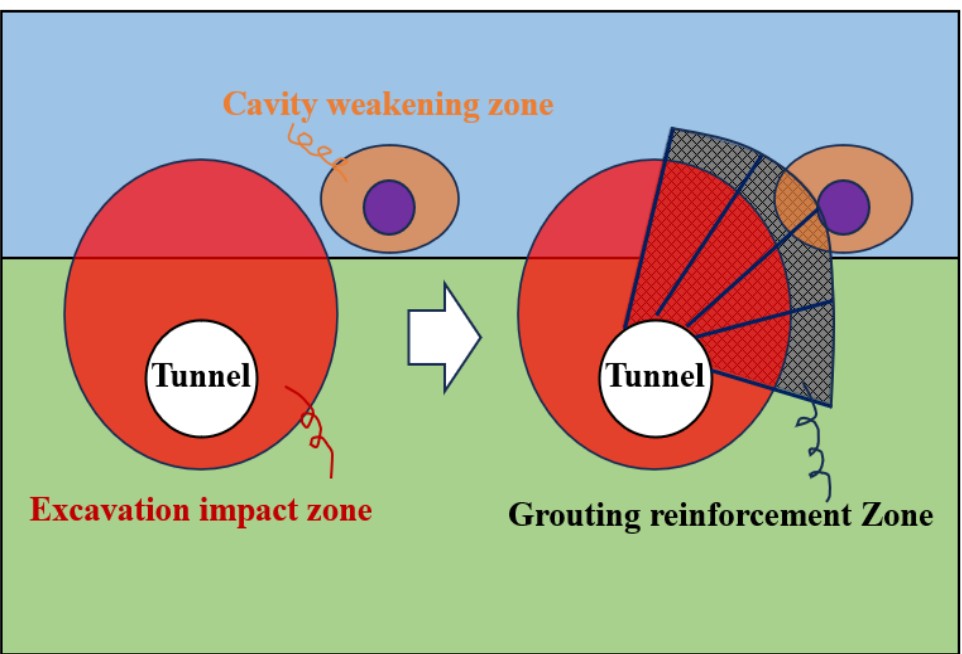

**Figure 16.** Reinforcement measures for strata containing the cavity.

Some limitations of this paper need to be stated. First, groundwater is an extremely dangerous factor for underground engineering. Groundwater causes seepage-coupled damage to formations by changing effective stresses, permeability, and mechanical properties. In particular, changes in pore water pressure lead to a reduction in the effective stress of the composite strata [20]. Future modifications can be made based on this test setup, aiming to study the failure mechanism of composite strata containing voids under the action of groundwater. In addition, cavities in strata are commonly present and there is uncertainty about their location, size, and number. Thus, it is recommended to systematically reveal the mechanism of the influence of the location, size, and number of cavities on the strata stability. In addition, the application of numerical methods such as discrete elements can further reveal the collapse mechanism of complex strata in the future.

## 5. Conclusions

In this paper, the deformation law and progressive collapse of composite strata with the cavity induced by tunnel excavation were investigated using indoor tests. A total of six groups of tests were designed based on the location of the cavity and the distance of the cavity from the tunnel. Loading was carried out at the top of the strata to investigate the effect of the cavity on the failure process of the composite strata. The following main conclusions were obtained:

(1) The strata without cavities had good stability after tunnel excavation. There was no obvious change inside the tunnel at the beginning of the upper load application. With the increase in the upper load, a small arch collapse first appeared, and the strata appeared to collapse when the upper load was increased to 10 kPa. When there was a cavity in the ground, the strata around the cavity were the first to be damaged by the upper loads applied after the tunnel was excavated. When the tunnel broke down into the overlying soil layer, the strata collapse progressed rapidly to a complete collapse.

(2) The stability of tunnels with a cavity located in different position conditions was studied. The existence of a cavity increased the ground deformation, and the cavity at different locations affected the ground deformation. When the cavity was located diagonally above the tunnel and on the outside of the tunnel foot, the surface settlement caused by the tunnel excavation was greater and the ground damage process was faster than when the cavity was located directly above the tunnel. The surface subsidence trough caused by the diagonally upper cavity at the soil–rock composite stratigraphic interface was larger than that at other locations. The location of the cavity changed the stress distribution state of the strata and thus altered the emergence of cracks, which finally changed the collapse process and pattern of the composite strata.

(3) With the increased distance between the tunnel and the cavity, the maximum surface settlement value decreased accordingly, and the intensity of the strata failure was also reduced accordingly. The distance between the tunnel and the cavity was 2 m, 4 m, and 6 m, the maximum deformation of the ground surface was 3.42 mm, 3.16 mm, and 3 mm, and the bearing capacity was 4 kPa, 6 kPa, and 8 kPa, respectively. The probability and range of stratum collapse increased when the tunnel excavation impact zone was connected with the cavity weakening zone. Therefore, a reinforcement measure for strata containing a cavity was proposed.

**Author Contributions:** Methodology, R.Z., Y.Z., S.L., Y.G. and S.Z.; Validation, Y.G.; Formal analysis, S.L.; Investigation, R.Z., Y.Z. and S.L.; Resources, S.Z.; Data curation, R.Z., Y.Z., S.L. and Y.G.; Writing—original draft, R.Z., Y.Z. and Y.G.; Writing—review & editing, R.Z. and S.Z.; Project administration, S.Z.; Funding acquisition, S.Z. All authors have read and agreed to the published version of the manuscript.

**Funding:** The authors gratefully acknowledge the support of the National Natural Science Foundation of China (No. 51978356) and the Demonstration Project of Benefiting People with Science and Technology of Qingdao, China (No. 23-2-8-cspz-13-nsh).

**Data Availability Statement:** The original contributions presented in the study are included in the article, further inquiries can be directed to the corresponding author.

**Conflicts of Interest:** Author Ruichuan Zhao was employed by the company CCCC HIGHWAY CONSULTANTS Co., Ltd. Author Yongjian Guo was employed by the company Qingdao Conson Second Jiaozhou Bay Subsea Tunnel Co., Ltd. The remaining authors declare that the research was conducted in the absence of any commercial or financial relationships that could be construed as a potential conflict of interest.

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
