# Peer review of "Experiments Studying the Instability Process of a Subway Tunnel in Soil–Rock Composite Strata Influenced by Defects"

_buildings, doi:10.3390/buildings15060878_

Round 1

Reviewer 1 Report

Comments and Suggestions for Authors

This study investigates the instability mechanisms of subway tunnels in soil-rock composite strata with inner cavities through systematic indoor experiments. The research addresses a critical gap in understanding defect-tunnel interactions, particularly the role of cavity location and distance in triggering ground deformation and collapse. The experimental design, incorporating high-resolution monitoring and controlled excavation simulations, provides valuable insights into progressive failure patterns. However, the study has notable limitations in theoretical validation, model representativeness, and contextualization with existing literature. The following are the specific comments:

  1. The manuscript lacks a detailed theoretical background on how inner cavities influence tunnel stability in soil-rock composite strata. A more comprehensive literature review should be included, focusing on the mechanical modeling of defect-induced instability.
  2. The similarity ratios for stress (Ce=20) and displacementCπ=20) are not explicitly justified. Provide a detailed derivation of scaling laws, ensuring dimensional consistency across all parameters (e.g., gravity Cγ= 1 may conflict with stress scaling).
  3. Deflating airbags to simulate cavities may not replicate natural void geometries (e.g., irregular shapes, stress history). Discuss limitations and consider alternative techniques (e.g., 3D-printed voids).
  4. The 1:20 scaling ratio may oversimplify field conditions. Discuss scaling distortions (e.g., gravity effects) and their implications for real-world applications.
  5. The manuscript does not address the role of groundwater, which could exacerbate failure in cavities. Including a discussion on how water ingress may influence the stability of tunnels in these strata, especially in terms of pore pressure and seepage erosion, would provide a more holistic view of the problem.It is recommended to cite the following reference (https://doi.org/10.1016/j.compgeo.2024.106944), which provides a theoretical framework for modeling transport processes in heterogeneous media. This framework can be applied to groundwater or other fluid flow systems surrounding tunnels. Incorporating this work will strengthen the manuscript’s discussion on how inner cavities and groundwater influence tunnel stability.
  6. The current analysis could be expanded to include a more comprehensive evaluation of how different cavity locations influence deformation patterns and failure mechanisms. Strengthening this discussion would enhance the manuscript’s contribution to understanding strata instability in complex geological conditions.
  7. Proposed grouting measures (Fig.15) are inadequately detailed. Specify grout material properties, injection pressures, and validation tests.
  8. Key figures (e.g., Figs.5-9) lack scale bars or timestamps for failure progression. Enhance visualization to clarify spatiotemporal dynamics.
  9. While cavity location is discussed in detail, the manuscript does not address the impact of cavity size on tunnel stability. Including tests with varying cavity sizes could provide more insight into the threshold beyond which the stability of the tunnel is compromised.
  10. The manuscript would benefit from more precise language and a clearer structure in some sections. Reducing redundancy and enhancing the flow of information would make the paper more accessible to a wider audience.
  11. The manuscript still contains minor writing issues, such as the unnecessary period at the end of the title.

Author Response

Response to Reviewer 1 Comments

1. Summary

2. Questions for General Evaluation

Reviewer’s Evaluation

Response and Revisions

Does the introduction provide sufficient background and include all relevant references?

Can be improved

Modifications have been made to the introduction.

Are all the cited references relevant to the research?

Can be improved

The cited references are relevant to the research.

Is the research design appropriate?

Can be improved

The framework and design of the research has been refined.

Are the methods adequately described?

Can be improved

A detailed description of the research methodology has been added.

Are the results clearly presented?

Can be improved

A clearer description of the results has been provided.

Are the conclusions supported by the results?

Can be improved

The conclusion has been re-polished.

3. Point-by-point response to Comments and Suggestions for Authors

Comments 1: The manuscript lacks a detailed theoretical background on how inner cavities influence tunnel stability in soil-rock composite strata. A more comprehensive literature review should be included, focusing on the mechanical modeling of defect-induced instability.

Response 1: Thanks for the reviewer’s valuable comment. Cavities and fractures, whether pre-existing or induced by excavation, disrupt stress arching and reduce load-bearing capacity. Previous work has pointed out that the cavity in composite strata significantly alters mechanical responses, necessitating targeted experimental studies. We have conducted more comprehensive literature research focusing on the mechanical modeling of defect-induced instability. The introduction has been supplemented with more detailed and specific descriptions.

Comments 2: The similarity ratios for stress (Ce=20) and displacementCπ=20) are not explicitly justified. Provide a detailed derivation of scaling laws, ensuring dimensional consistency across all parameters (e.g., gravity Cγ= 1 may conflict with stress scaling).

Response 2: We sincerely appreciate the reviewer’s insightful feedback regarding the justification of similarity ratios and dimensional consistency. Below is a detailed derivation of the scaling laws and their validation: Similarity theory dictates that the equilibrium equations, geometric compatibility conditions, constitutive relations, stress boundaries, and displacement boundaries must satisfy strict congruence between prototype and scaled models. The similarity ratios of the model test have the following relation:

                                     (1)

where Cσ, CL, Cγ, Cδ, Cε, and CE represent the similarity ratio of stress, geometry, density, displacement, strain, and elastic modulus, respectively. The derivation of scaling laws has been added in section 2.1 of the resubmission.

Comments 3: Deflating airbags to simulate cavities may not replicate natural void geometries (e.g., irregular shapes, stress history). Discuss limitations and consider alternative techniques (e.g., 3D-printed voids).

Response 3: Thank you for raising this point regarding the limitations of airbag deflation in cavity simulation. We fully agree that the geometric and mechanical discrepancies between artificial airbag voids and natural cavities require careful consideration. While airbag methods simplify cavity simulation by generating axisymmetric voids, they inadequately capture the irregular morphologies. However, it has been proven through established research that airbags to simulate cavities are a simple and effective method. This simplified method can also better represent the changes in the strata stability caused by cavities [1-3]. Therefore, considering the feasibility and economy, this paper also chooses the airbag to simulate the cavity. As the reviewer suggests, 3D printing technology opens up new opportunities for indoor experiments with its advantages in producing complex models. Therefore, the future application of 3D printing technology to our research is to be expected. The limitations of deflating airbags have been added to the resubmission. Once again, we thank the reviewer for their valuable comments.

[1]      Zhang, C.; Gao, J.; Wang, Z.; Liu, C. Model Test on the Collapse Evolution Law of Tunnel Excavation in Composite Strata with a Cavity. 2024.

[2]      Cai, Y.; Zhang, C.; Min, B. Deformation and Failure Characteristics of Strata Induced by the Construction of Shallow Tunnels Adjacent to Overlying Voids. Journal of the China Railway Society 2019, 41(9), 118-127.

[3]      Yang, G. Interaction Mechanism between Shallow Tunnels and Adjacent Stratum Voids and Its Impact on Ground Deformation [Ph.D. Thesis, Beijing Jiaotong University]. 2021. DOI:10.26944/d.cnki.gbfju.2021.000324.

Comments 4: The 1:20 scaling ratio may oversimplify field conditions. Discuss scaling distortions (e.g., gravity effects) and their implications for real-world applications.

Response 4: Thanks for the reviewer's valuable comment. We appreciate the reviewer’s valid concern regarding potential scaling distortions in the 1:20 physical model. In tunneling and geotechnical scaling model tests, failure to reasonably consider the multi-physical field relationships in the similarity theory may lead to significant deviations between the test results and the actual engineering behavior. The core reason for this deviation is that the gravity effect is not fully satisfied, and the simplification of material properties can further amplify the error. Therefore, the selection of appropriate similar materials is of great importance. The gypsum-based materials, sand-paraffin/resin, and barite powder-bentonite blends are similar materials that are often used, and 3D-printed composites are currently more popular. However, all of the above materials have advantages and limitations and need to be employed according to the research objectives and requirements. Considering the properties, ease of access and economics of similar materials, barite, quartz sand and petroleum jelly were selected in conjunction with established research. As the reviewer suggests, the limitations of the scaling model tests have been added in the resubmission.

Comments 5: The manuscript does not address the role of groundwater, which could exacerbate failure in cavities. Including a discussion on how water ingress may influence the stability of tunnels in these strata, especially in terms of pore pressure and seepage erosion, would provide a more holistic view of the problem. It is recommended to cite the following reference (https://doi.org/10.1016/j.compgeo.2024.106944), which provides a theoretical framework for modeling transport processes in heterogeneous media. This framework can be applied to groundwater or other fluid flow systems surrounding tunnels. Incorporating this work will strengthen the manuscript’s discussion on how inner cavities and groundwater influence tunnel stability.

Response 5: The authors are very grateful to the reviewer's suggestion. Groundwater is an extremely dangerous factor for underground engineering. Groundwater causes seepage-coupled damage to formations by changing effective stresses, permeability, and mechanical properties. In particular, changes in pore water pressure lead to a reduction in the effective stress of the formation, which reduces the strength of the soil; infiltration damage caused by dynamic water pressure, such as pipe surges; and softening or swelling of the soil due to changes in the water table. Combined with the excellent literature recommended, the impact of groundwater on the stability of strata has been analyzed. Incorporating the reviewer's recommendations, future modifications can be made based on this test setup, aiming to study the failure mechanism of composite strata containing voids under the action of groundwater. Based on the reviewer's suggestion, the above descriptions have been added to the resubmission.

Comments 6: The current analysis could be expanded to include a more comprehensive evaluation of how different cavity locations influence deformation patterns and failure mechanisms. Strengthening this discussion would enhance the manuscript’s contribution to understanding strata instability in complex geological conditions.

Response 6: The authors are very grateful to the reviewer's suggestion. We sincerely appreciate the reviewer's constructive suggestion to strengthen the analysis of cavity location impacts. Based on the reviewers' suggestions, we further examined the progressive collapse of composite strata influenced by a cavity at different locations. The extension process and evolutionary characteristics of cracks during the loading process are added. It is found that the location of voids changed the stress distribution state of the formation, which in turn changed the appearance of cracks. In addition, the location of voids is found to affect the collapse process and mode of composite strata. A description of the collapse mechanism and process has been added to further emphasize the research focus. The revisions are emphasized in the resubmission.

Comments 7: Proposed grouting measures (Fig.15) are inadequately detailed. Specify grout material properties, injection pressures, and validation tests.

Response 7: The authors are very grateful to the reviewer's suggestion. Grouting reinforcement of subway tunnel strata is a key technology to enhance the strata stability. The following commonly used common grouting materials include cement slurry, cement-water-glass bi-liquid slurry, chemical slurry, and ultra-fine cement slurry. The material chosen needs to be decided according to the strata type. The grouting pressure is usually 0.5-3.0 MPa, depending on the formation conditions and the depth of the grouting. The initial grouting pressure is low and gradually increases to the design pressure. Of course, the effectiveness and reasonableness of the grouting material and grouting pressure can be validated by indoor tests and field tests. Therefore, through a reasonable selection of materials and processes, grouting reinforcement can effectively improve the stability and safety of subway tunnels. Thanks for the reviewer's suggestion, the descriptions have been added to the resubmission.

Comments 8: Key figures (e.g., Figs.5-9) lack scale bars or timestamps for failure progression. Enhance visualization to clarify spatiotemporal dynamics.

Response 8: The authors are very grateful to the reviewer's suggestion. Apologize for our carelessness. The description of representative pictures of each stage has been added. Based on this, a further in-depth analysis of the progressive collapse process has been developed to highlight the characteristics of each stage. The revisions are highlighted in the resubmission.

Comments 9: While cavity location is discussed in detail, the manuscript does not address the impact of cavity size on tunnel stability. Including tests with varying cavity sizes could provide more insight into the threshold beyond which the stability of the tunnel is compromised.

Response 9: We sincerely appreciate the reviewer’s insightful suggestion regarding the critical role of cavity size effects in tunnel stability. Cavities in strata are commonly present and there is uncertainty about their location and size. This paper focuses on the location of cavities on strata stability and figures out the mechanism of their influence. As the reviewer pointed out, the size of cavities affects the stability of strata. Thus, we will conduct further research in the future to systematically reveal the mechanism of the influence of the size of cavities on strata stability. Taking into account the reviewer's valuable suggestion, we have emphasized this point in the outlook, which will be a subsequent task. We would like to thank the reviewers for their valuable suggestion, which have helped us to further deepen our understanding of this topic.

Comments 10: The manuscript would benefit from more precise language and a clearer structure in some sections. Reducing redundancy and enhancing the flow of information would make the paper more accessible to a wider audience.

Response 10: Thanks for the reviewer's valuable comment. We are deeply sorry for the trouble caused to you by our redundancy language. In the introduction of the resubmitted paper, the language and structure were reoptimized. We have modified the corresponding sections to be more accurate and clearer.

Comments 11: The manuscript still contains minor writing issues, such as the unnecessary period at the end of the title.

Response 11: Thanks for the reviewer's valuable comment. We are deeply sorry for the trouble caused to you by our writing issues. We have modified the corresponding writing issues to be more accurate and clearer. Thanks again.

4. Response to Comments on the Quality of English Language

Point 1: The English is fine and does not require any improvement.

Response 1: The quality of the English language is polished to meet publication requirements.

5. Additional clarifications

/

Reviewer 2 Report

Comments and Suggestions for Authors

This article discusses the instability process of subway tunnels in soil-rock composite strata under the influence of defects using indoor experimental methods. The study is particularly concerned with how cavities affect tunnel stability during excavation and under load. Six groups of tests were carried out, varying the location and distance of the cavity from the tunnel. The findings verify that cavities have significant effects on ground deformation and increase collapse hazards, particularly if they are located in near-field position of the excavation influence area. The authors propose that the findings in this paper can contribute to safer tunnel excavations by providing improvement alternatives in tunnel construction activities like reinforcement through grouting. The paper is interesting and can be published provided the following issues of the reviewers and editor are addressed by the authors:

The paper is helpful in giving experimental evidence about the instability of tunnels in composite strata made of soil-rock. However, the novelty of the work has to be made more apparent. How does the research advance the state-of-the-art over past research?

The use of a similarity model is logical, but additional details on the model validation are needed. How closely do the results from the scaled-down models compare to actual field conditions? Providing references or comparisons to real projects would make the study more applicable.

The selection of material properties for the physical model is logical, but were there other materials considered as alternatives? Discussion of the limitations of the chosen materials in replicating actual strata behavior would be useful.

The paper describes highly detailed experimental results, yet the discussion may be lengthened by including more detailed theoretical consideration. Are observed failure modes in comparison with present numerical or analytical models?

The literature review needs more depth. To better position your work, provide a detailed analysis and synthesis of relevant studies. Currently, it is too brief. Review the following papers on rock strength prediction:
http://dx.doi.org/10.55579/jaec.202154.344;
https://doi.org/10.3390/buildings14051236

The phrase "The findings of the study are hoped to provide lessons..." is somewhat vague. A more definitive statement, such as "The findings provide critical insights for..." would be stronger.

The results suggest that cavities influence failure modes, but can the authors elaborate on the dominant failure mode observed (e.g., shear failure, tensile cracking, progressive collapse)?

How does groundwater presence affect the observed failure patterns, and how could this be incorporated into future studies?

Given that the model box has physical constraints, could the size of the test box introduce boundary effects that alter stress distribution and failure mechanisms?

What measures were taken to minimize artificial boundary influences?

How would larger cavities or multiple closely spaced cavities affect the stability of the tunnel?

Author Response

Response to Reviewer 2 Comments

1. Summary

2. Questions for General Evaluation

Reviewer’s Evaluation

Response and Revisions

Does the introduction provide sufficient background and include all relevant references?

Can be improved

Modifications have been made to the introduction.

Are all the cited references relevant to the research?

Can be improved

The cited references are relevant to the research.

Is the research design appropriate?

Can be improved

The framework and design of the research has been refined.

Are the methods adequately described?

Can be improved

A detailed description of the research methodology has been added.

Are the results clearly presented?

Can be improved

A clearer description of the results has been provided.

Are the conclusions supported by the results?

Yes

Thanks a lot.

3. Point-by-point response to Comments and Suggestions for Authors

Comments 1: The paper is helpful in giving experimental evidence about the instability of tunnels in composite strata made of soil-rock. However, the novelty of the work has to be made more apparent. How does the research advance the state-of-the-art over past research?

Response 1: The authors are very grateful to the reviewer's suggestion. The study is particularly concerned with how cavities affect tunnel stability during excavation and under load. Moreover, the influence of the location of the cavity was figured out, which is an important guide for the stability control of composite strata with cavity. The previous studies on collapse in composite strata are further summarized, and it is found that the existing studies mainly focus on the collapse process of homogeneous strata and the effect of voids on the mechanical response of the strata. However, there are few studies on the collapse mechanism of soil-rock composite strata influenced by the location of the cavity. Combining the reviewer's valuable suggestion, the novelty, and state-of-the-art of this work has been repolished in the introduction. The related modifications are highlighted in the resubmission.

Comments 2: The use of a similarity model is logical, but additional details on the model validation are needed. How closely do the results from the scaled-down models compare to actual field conditions? Providing references or comparisons to real projects would make the study more applicable.

Response 2: The authors are very grateful to the reviewer's suggestion. At present, collapse always happens when tunneling in composite strata. Therefore, a similar model test is adopted to study the collapse mechanism of composite strata with an inner cavity. Based on this, the influence of the cavities located in different positions on the stability of the strata is figured out. At present, the collapse of real projects is sudden and the causes of the collapse are very complicated. Therefore, it is very difficult to grasp the collapse evolution process for a real project. Combined with the reviewer's suggestion, several examples of actual engineering collapses are selected to compare the final collapse pattern of the composite strata (RFig.6). It can be found that the fracture surface of the strata in both the model test and the actual project is larger the closer it is to the subsurface. The final collapse patterns of both are the same. As the reviewer suggests, the comparisons to real projects have been added to section 3.1 in the resubmission.

RFig.6. Collapse pattern of composite strata [1].

[1]     Liu, C.; Zhang, S.; Zhang, D.; Zhang, K.; Wang, Z. Model Tests on Progressive Collapse Mechanism of a Shallow Subway Tunnel in Soft Upper and Hard Lower Composite Strata. Tunn. Undergr. Space Technol. 2023, 131, 104824. https://doi.org/10.1016/j.tust.2023.104824

Comments 3: The selection of material properties for the physical model is logical, but were there other materials considered as alternatives? Discussion of the limitations of the chosen materials in replicating actual strata behavior would be useful.

Response 3: We appreciate the reviewer's insightful comment regarding material selection.

In similar model tests for tunnels, the selection of similar materials for the surrounding rock needs to be based on similarity theory (geometric and mechanical ratios) to reproduce as much as possible the mechanical behavior of the actual strata (e.g., strength, deformation, etc.). In this study, the similar materials and their mix proportions employed were experimentally validated through laboratory tests, confirming their compliance with specified mechanical parameters. The gypsum-based materials, sand-paraffin/resin, and barite powder-bentonite blends are similar materials that are often used, and 3D-printed composites are currently more popular. However, all of the above materials have advantages and limitations and need to be employed according to the research objectives and requirements. Considering the properties, ease of access and economics of similar materials, barite, quartz sand and petroleum jelly were selected in conjunction with established research. The mixing of barite powder increases the brittleness of the material, which may lead to sudden brittle cracks in the model during the loading process. The sharp edges of artificial quartz sand lead to a higher friction angle in the model than that of the natural sand layer, which overestimates the self-supporting capacity of surrounding rock to a certain extent and may conceal the potential shear slip risk in the actual project. Petroleum jelly as a cementing agent could reduce the inter-particle occlusion force. Despite the limitations mentioned above, model tests are a very common research tool today. It has significant advantages in representing the real mechanical state and failure process of the strata. Therefore, it is important to define the research objectives, rationally select materials, and quantify their applicability when designing the test. Taking into account the reviewer's valuable comments, we have added an analysis of the limitations of similar materials in section 2.3 to facilitate the readers to make a rational choice for subsequent experiments. We would like to thank the reviewers for their valuable comments, which have given us a better understanding of the principles of using similar materials and the selection aspects.

Comments 4: The paper describes highly detailed experimental results, yet the discussion may be lengthened by including more detailed theoretical considerations. Are observed failure modes in comparison with present numerical or analytical models?

Response 4: The authors are very grateful to the reviewer's suggestion. Indoor tests, analytical models, and numerical simulations are commonly used research methods. In this paper, the indoor test is adopted to study the progress collapse of composite strata containing cavities. It is found that the strata collapse is affected by material properties and the loading process and has obvious discontinuity and uncertainty. It is well known that the analytical model has significant advantages in studying intrinsic mechanisms. However, the analytical model generally requires basic assumptions such as homogenization and continuity, which are not suitable for large deformation and failure processes in strata. In addition, the finite element, the finite difference, and the finite discrete element are frequently used in numerical simulation methods. The finite element is suitable for the analysis of continuous medium, while the discrete element and finite discrete element need to make great efforts in parameter calibration, and the parameter calibration process has a direct impact on the calculation results. Liu et al (2023) adopted the DEM to investigate the collapse process of the composite strata. Although the designed working conditions are different (thickness of rock layer, mechanical parameters of materials), it can be seen that this failure pattern is somewhat similar. At present, a large number of analytical models and numerical simulations have been developed for homogeneous strata. However, there are few analytical models and numerical simulations for cavity-containing strata. Taking the reviewers' suggestions into account, we have further enriched the description of the experimental results. In the future, we will try to reproduce the process by numerical simulation, and we thank the reviewers for their valuable suggestions.

RFig Comparison of collapse mode with numerical simulations (Two stages)

[1]       Liu, C.; Zhang, S.; Zhang, D.; Zhang, K.; Wang, Z. Model Tests on Progressive Collapse Mechanism of a Shallow Subway Tunnel in Soft Upper and Hard Lower Composite Strata. Tunn. Undergr. Space Technol. 2023, 131, 104824. https://doi.org/10.1016/j.tust.2023.104824

Comments 5: The literature review needs more depth. To better position your work, provide a detailed analysis and synthesis of relevant studies. Currently, it is too brief. Review the following papers on rock strength prediction:

http://dx.doi.org/10.55579/jaec.202154.344;

https://doi.org/10.3390/buildings14051236.

Response 5: Thanks for the reviewer's valuable comment. A further summary of the current state of research on modeling tests for tunnel collapse is given in the introduction. The importance of defects is highlighted, and the impact of ground defects on the safety of tunnel construction is pointed out. The research on the stability of composite strata is reorganized by combining the important literature [1,2] recommended by the reviewers. The research highlights of this paper are polished through the above organization. The modifications have been highlighted in the resubmission.

[1]       Rezamand, A.; Afrazi, M.; Shahidikhah, M. Study of Convex Corners’ Effect on the Displacements Induced by Soil-Nailed Excavations. JOURNAL Adv. Eng. Comput. 2021, 277–290. http://dx.doi.org/10.55579/jaec.202154.344

[2]       Wu, X.; Xu, J.; Wang, S.; Sha, P.; Han, Z.; Chen, X.; Shu, S.; Qiao, W. Ground Deformation of Shield Tunneling through Composite Strata in Coastal Areas. 2024. https://doi.org/10.3390/buildings14051236

Comments 6: The phrase "The findings of the study are hoped to provide lessons..." is somewhat vague. A more definitive statement, such as "The findings provide critical insights for..." would be stronger.

Response 6: The authors are very grateful to the reviewer's suggestion. In this paper, the influence of different locations of cavities on the stability of composite strata is revealed through modeling tests. The obtained conclusions can provide technical support for the collapse prevention and safety control of composite strata containing cavities. We have modified the corresponding statement to be stronger. The resubmission has been revised as follows: "The findings can provide technical support for the collapse prevention and safety control of subway tunnel in composite strata with internal defects."

Comments 7: The results suggest that cavities influence failure modes, but can the authors elaborate on the dominant failure mode observed (e.g., shear failure, tensile cracking, progressive collapse)?

Response 7: The authors are very grateful to the reviewer's suggestion. Combining with the reviewer's suggestion, we further polish the progressive collapse of the composite strata under the influence of voids. The extension process of cracks under loading and the evolutionary characteristics are added, and it is found that the location of the voids changed the stress distribution state of the strata and thus altered the emergence of cracks. In addition, it is also found that the location of the voids affects the collapse process and pattern of the composite strata. Therefore, according to the reviewer's suggestion, we added a description of the collapse mechanism and process to further emphasize the research focus of this paper. The modifications have been highlighted in the resubmission.

Comments 8: How does groundwater presence affect the observed failure patterns, and how could this be incorporated into future studies?

Response 8: The authors are very grateful to the reviewer's suggestion. Groundwater is an extremely dangerous factor for underground engineering. Groundwater causes seepage-coupled damage to formations by changing effective stresses, permeability, and mechanical properties. In particular, changes in pore water pressure lead to a reduction in the effective stress of the formation, which reduces the strength of the soil; infiltration damage caused by dynamic water pressure, such as pipe surges; and softening or swelling of the soil due to changes in the water table. Future modifications can be made based on this test setup, aiming to study the failure mechanism of composite strata containing voids under the action of groundwater. Based on the reviewer's suggestion, the above descriptions have been added to the resubmission.

Comments 9: Given that the model box has physical constraints, could the size of the test box introduce boundary effects that alter stress distribution and failure mechanisms? What measures were taken to minimize artificial boundary influences?

Response 9: Thanks for the reviewer's valuable comment. We apologize for not introducing the detailed process in the paper. It is shown that when the distance of the model boundary from the tunnel center is more than 3 times the tunnel diameter, the influence of the boundary effect is not significant. The scale of the model test is 1:20, the tunnel diameter is 30cm, and the tunnel center is 100cm away from the model boundary (about 3.3 times the tunnel diameter). Therefore, the influence of the boundary effect in this experiment is not significant. In addition, to further weaken the boundary effect, the bottom and both sides of the test box were filled with cushioning material with similar stiffness to the soil body. Moreover, this experiment is a staged application of load at the top of the model to avoid the superposition of stress waves reflected at the boundary due to transient loading. Therefore, the above measures were used to mitigate the boundary effect in this experiment, and the corresponding description has been added in section 2.4 of the resubmission.

Comments 10: How would larger cavities or multiple closely spaced cavities affect the stability of the tunnel?

Response 10: We sincerely appreciate the reviewer’s insightful suggestion regarding the critical role of cavity size and multiple closely spaced cavities effects in tunnel stability. Cavities in strata are commonly present and there is uncertainty about their location, size, and number. This paper focuses on the location of cavities on strata stability and figures out the mechanism of their influence. As the reviewer pointed out, the location, size, and number of cavities affect the stability of strata. Thus, future research should be conducted to systematically reveal the mechanism of the influence of the location, size, and number of cavities on the strata stability. Taking into account the reviewer's valuable suggestion, we have emphasized this point in the outlook, which will be a subsequent task. We would like to thank the reviewers for their valuable suggestion, which have helped us to further deepen our understanding of this topic.

4. Response to Comments on the Quality of English Language

Point 1: The English is fine and does not require any improvement.

Response 1: The quality of the English language is polished to meet publication requirements.

5. Additional clarifications

/

Round 2

Reviewer 1 Report

Comments and Suggestions for Authors

Based on the substantial improvements made by the authors in addressing identified methodological and analytical deficiencies during the revision process, I recommend acceptance for publication in its current form.

Reviewer 2 Report

Comments and Suggestions for Authors

Accept.